# Imaging strategies for follow-up during adjuvant nivolumab in esophageal cancer: A multicenter retrospective cohort study

Tamara J. Huizer[1☉], Anniek Strijdhorst [2,3,4☉]*, Laurens V. Beerepoot[5],
Mark I. van Berge Henegouwen[4,6], Leni van Doorn[1], Sarah Derks[4,7,8], Bas P. L. Wijnhoven[9],
Bianca Mostert[1☉], Hanneke W. M. van Laarhoven[2,4☉]

1 Department of Medical Oncology, Erasmus MC Cancer Institute University Medical Center, Rotterdam, the Netherlands, 2 Department of Medical Oncology, Amsterdam UMC, Location University of Amsterdam, Amsterdam, the Netherlands, 3 Amsterdam Cardiovascular Sciences, Atherosclerosis & Ischemic Syndromes, Amsterdam, the Netherlands, 4 Cancer Center Amsterdam, Cancer Treatment and Quality of Life, Amsterdam, the Netherlands, 5 Department of Medical Oncology, Elisabeth Tweesteden Hospital, Tilburg, the Netherlands, 6 Department of Surgery, Amsterdam UMC, Location University of Amsterdam, Amsterdam, the Netherlands, 7 Department of Medical Oncology, Amsterdam UMC, Location Free University, Amsterdam, the Netherlands, 8 Oncode Institute, Utrecht, the Netherlands, 9 Department of Surgery, Erasmus MC Cancer Institute University Medical Center, Rotterdam, the Netherlands

☉ These authors contributed equally to this work.
* a.strijdhorst@amsterdamumc.nl

## Abstract

### Introduction

Adjuvant nivolumab improves disease-free survival in patients with esophageal or gastroesophageal junction cancer following neoadjuvant chemoradiotherapy and resection, but recurrence risk remains high. Optimal follow-up strategies for early detection of recurrence during nivolumab treatment are lacking.

### Methods

In this multi-center, retrospective cohort study, patients from three hospitals in the Netherlands who underwent surgical resection between September 2021 and October 2023 and received adjuvant nivolumab were included. Data on tumor type, treatment, and imaging intervals (every 3 months vs every 4 months) were collected. Primary outcome was disease recurrence, secondary outcomes included disease-free survival (DFS), and diagnostic yield of CT scans.

### Results

Of 353 patients that underwent resection, 151 received at least one dose of adjuvant nivolumab (82% male; median age 66 years; 86% adenocarcinoma). A total of 27 patients (18%) developed on-treatment recurrent disease, which was detected by routinely performed CT scans in 67% and included distant metastasis in 89%. DFS

**Data availability statement:** Data cannot be shared publicly due to ethical restrictions related to participant consent. The ethics approval (MEC2023-0631) specified that individual-level data would not be publicly released. Requests for de identified, aggregated data (including summary statistics and additional subgroup analyses upon request) may be directed to the Department of Internal Oncology at Erasmus University Medical Center via interne. oncologie@erasmusmc.nl and will require approval from the Medical Research Ethical Committee of Erasmus University Medical Center Rotterdam.

**Funding:** The author(s) received no specific funding for this work.

**Competing interests:** The authors declare that they have no known competing financial interests or personal relationships that could have appeared to influence the work reported in this paper. Disclaimer: TJH and AS have no discolures. HvL reports: Research funding and/or medication supply: Amphera, Anocca, Astellas, AstraZeneca, Beigene, Boehringer, BMS,Daiichy-Sankyo, Dragonfly, MSD, Myeloid, ORCA, Servier; Consultant/advisory role: Auristone, Incyte, Merck, Myeloid, Servier; Speaker role: Astellas, Beigene, Benecke, BMS, Daiichy-Sankyo, JAAP, Medtalks, Novartis, Springer, and Travel Congress Management B.V. BM reports: Research funding and/or medication supply: BMS, Pfizer; Consultant/advisory role: Lilly, AstraZeneca; Speaker role: Servier, BMS, Amgen. SD reports: a consultant or advisory role for BMS (related to checkpoint inhibitors); research funding, medication supply, or both from Incyte (related to checkpoint inhibitors); and speaker roles for Servier, BMS, and Benecke. LVB reports: speaker role: Medtalks, BMS, Servier, Travel Congress Management. BW reports: Research funding BMS, consulting and speaker fee Medtronic, speaker role Travel Congress Management B.V. MvBH declares consultancies for Johnson and Johnson, Stryker, BBraun Intuitive and Medtronic. All fees and grants paid to institution.

was 89% at 4 months, 84% at 8 months, and 75% at 12 months, showing a gradual decline over time. The diagnostic yield of CT scans increased from 0% at baseline to 9% at 4 months.

## Conclusion

Routine baseline CT imaging did not detect recurrences, while routine imaging during adjuvant nivolumab identified the majority of recurrences. The gradual decline in disease-free survival suggests that recurrences are distributed over time, supporting a routine imaging interval, such as every 3 or 4 months as used in our study. These real-world data may help inform clinicians, and future studies can further evaluate optimal imaging intervals.

## Introduction

Esophageal cancer is the eighth most commonly diagnosed cancer and the sixth leading cause of cancer death worldwide, with a 5-year survival as low as 30% [1–4]. Neoadjuvant chemoradiotherapy (nCRT) followed by surgery is a well-established standard of care for resectable, locally advanced esophageal or gastroesophageal junction cancer [5]. However, even with curative intent, 50% of patients develop distant metastasis within 5 years [6–9]. Until 2021, clinical follow-up was the standard of care following nCRT and surgery.

In recent years, immune checkpoint inhibitors (ICI) have shown significant potential in the treatment of a wide range of cancers, including those that have historically been difficult to manage. The Checkmate 577 trial demonstrated that adjuvant nivolumab (anti-programmed cell death protein 1 antibody), administered for up to one year or until recurrence, significantly increased disease-free survival (DFS) in patients with residual pathological disease after microscopic radical resection (R0) resection compared to placebo (22.4 vs 11.0 months) [10]. Based on these results, the European Medicines Agency (EMA) approved adjuvant nivolumab in 2021 for this indication [5,11].

Updated results from CheckMate 577, presented at the 2025 American Society of Clinical Oncology (ASCO) Annual Meeting, showed a persistent DFS benefit. However, while a trend towards overall survival (OS) was observed (median 51.7 vs 35.3 months; HR 0.85, P = 0.1064). These data have been presented in abstract form only and have not yet undergone peer review. Final conclusions regarding overall survival benefit should await full publication [12].

Despite improved DFS, approximately 38% of patients treated with adjuvant nivolumab in the Checkmate 577 developed local and/or metastatic recurrent disease. This underscores the need for an effective follow-up strategy to avoid unnecessary exposure to ICI treatment, minimize toxicities and costs, and optimize healthcare resources, including limiting radiation exposure from computed tomography (CT) scans. In clinical practice, follow-up during adjuvant ICI is often performed before treatment initiation and every 3–4 months thereafter using CT scans, as

recommended in the Checkmate 577. Studies in other cancers, such as melanoma, suggest that follow-up frequency can be safely reduced based on recurrence patterns, potentially decreasing the number of CT scans without compromising detection rates [13]. This study aimed to evaluate the timing and detection of disease recurrence during adjuvant nivolumab, in order to inform optimal imaging intervals and follow-up strategies.

## Methods

### Subjects and study design

This was a retrospective cohort study performed in one teaching and two academic high-volume esophageal cancer centers in the Netherlands: Amsterdam University Medical Center (Amsterdam), Erasmus Medical Center (Rotterdam), and Elisabeth-TweeSteden hospital (Tilburg). All patients with a confirmed diagnosis of esophageal or gastroesophageal cancer who underwent surgery between September 1, 2021, and October 31, 2023, were identified from institutional databases.

Patients were assessed for eligibility if they matched the Checkmate 577 criteria, including neoadjuvant treatment with chemoradiotherapy prior to surgery, microscopic radical resection (R0), incomplete pathological response and who received at least one cycle of adjuvant nivolumab [14]. Patients with missing data on treatment initiation, recurrent disease and/or details of routine performed CT scans during adjuvant treatment were excluded.

Data collection and reporting followed the Strengthening the Reporting of Observational Studies in Epidemiology (STROBE) guidelines [15]. A completed STROBE checklist is provided as Supporting Information (S1 Checklist). Baseline demographic and clinical data were extracted from electronic health records, including age, sex, tumor characteristics (histology, stage), neoadjuvant treatment, surgery, adjuvant treatment, follow-up strategies, recurrent disease and method of detection, cancer-specific death, and death of any cause. When available, the CPS (Combined Positive Score) determined from pre-treatment biopsies or surgical specimens was extracted from patient records. The score was assessed in accordance with local practice and clinical care guidelines as part of standard care when indicated. The CPS was defined as the ratio of the total number of Programmed Death Ligand-1 (PD-L1) positive cells (tumor cells, lymphocytes, and macrophages) to the total number of viable tumor cells, multiplied by 100. Patients were followed until death, loss to follow-up, or until the end of study on August 12, 2024. The study was conducted in accordance with the Declaration of Helsinki and approved by the Medical Research Ethical Committee of Erasmus University Medical Center Rotterdam (MEC2023−0631). Informed consent was waived.

The primary outcome was disease recurrence and was defined as local, regional or distant from the primary resected site, confirmed by imaging, cytology and/or pathological evaluation. CT scans were performed as part of routine clinical care and were retrospectively reviewed for the purposes of this study. As such, board-certified radiologists interpreted the scans in accordance with standard clinical practice, and blinding was not applied. Recurrence was defined as the presence of new lesions suspicious for malignancy. In cases where clinical symptoms were later confirmed by imaging, the date of the CT scan was considered the date of recurrence. CT scans were classified as symptom-driven if the clinical indication was documented in the scan request or medical record, and the scan was performed outside the routine protocol.

Secondary outcomes included disease-free survival (DFS) and diagnostic yield of routine CT scans per time point. DFS, defined as the time (in months) from the start of adjuvant nivolumab to disease recurrence or death, whichever occurred first. Adjuvant nivolumab was initiated within 16 weeks after esophagectomy for every patient, per protocol. Unless otherwise specified, all timepoints and analyses are referenced from the start of adjuvant nivolumab.

The diagnostic yield of routine CT scans per time point, predictors associated with early (0–6 months) and late (7–12 months) recurrence, all-cause mortality, and comparison with DFS data from the CheckMate 577 were explored. Disease recurrence was routinely evaluated using CT imaging every 3 (Q3M) or 4 months (Q4M) from the first administration of nivolumab in addition to baseline and post-treatment imaging, according to local institutional standards. After completion

or discontinuation of adjuvant treatment, CT scans were performed based on clinical indications, including follow-up for toxicity, surgical monitoring, or clinical suspicion of disease recurrence.

### Statistical analysis

Descriptive statistics were used for baseline characteristics. Continuous variables were summarized as mean with standard deviation (SD) when normally distributed, or medians with interquartile ranges (IQR) for non-normally distributed variables. Categorical variables were summarized as counts and percentages. Disease recurrence was reported descriptively. DFS was calculated in months and estimated using the Kaplan-Meier method. Time-to-event curves were generated, with recurrence or death considered as events. Patients lost to follow-up were censored at their last known visit. The diagnostic yield per routine CT evaluation was calculated as the number of detected recurrences divided by the number of routine CT scans at each timepoint. Subanalysis was conducted to explore baseline predictors for early (0–6 months) and late (7–12 months) recurrence during treatment. To facilitate descriptive comparisons between this study and the CheckMate 577 trial, DFS data were extracted from the Kaplan-Meier curve of the CheckMate 577 trial using WebPlotDigitizer 4.1 software [16]. No sample size calculation was performed; all eligible patients between September 1, 2021 and October 31, 2023 were included. A two-sided P-value <0·05 was considered statistically significant. All statistical analyses were performed using R software version 4.3.2 (R Core Team (2018), R Foundation for Statistical Computing, Vienna, Austria).

## Results

### Cohort selection

A total of 451 patients underwent resection for esophageal or esophageal junction carcinoma between September 1, 2021, and October 31, 2023 (Fig 1). Of these, 98 patients (22%) were excluded based on treatment prior to surgery, including eighteen patients (5%) who participated in clinical trials with additional systemic therapy: tocilizumab in the BASALT trial (NCT04554771 [17]), trastuzumab/pertuzumab in the TRAP-2 trial (NCT05188313 [18]) and nivolumab in the SANO-3 study (NCT05491616 [19]).

Of the remaining 353 patients, 78% received nCRT with carboplatin/paclitaxel. After surgery, 74 patients (21%) had a complete pathological response, and 20 (6%) were excluded due to microscopic irradical resection (R1). Thirty patients (8%) were excluded due to missing data. A total of 229 (65%) patients were assessed for eligibility for adjuvant nivolumab, of whom 151 (43%) received at least one cycle.

The main reasons for not starting adjuvant treatment were poor performance status (n = 22), patient preference (n = 33), and symptomatic recurrence (n = 11) detected before starting adjuvant treatment.

### Nivolumab cohort

The nivolumab cohort included 151 patients, with a median age of 66 years (IQR, 60–72), 82% were male, and 86% had an adenocarcinoma (Table 1). The median number of completed nivolumab cycles was 10 (IQR, 5–13), with early discontinuation in 76 patients (50%), due to either immune-related adverse events (43%), recurrent disease (36%), patient preference (12%), or other reasons (9%) (Table 2). At study completion, twelve patients (8%) were still on treatment.

### Recurrent disease

During treatment, 27 (18%) (95% CI: 12–25%) patients developed recurrent disease, predominantly distant metastases (89%). Eighteen patients (67%) (95% CI: 46–84%) were asymptomatic and their recurrence was detected by routine imaging. DFS was assessed monthly during the first year after nivolumab initiation, with estimated rates of 89% (95% CI: 85–95%) at 4 months, 84% (95% CI: 79–90%) at 8 months, and 75% (95% CI: 69–83%) at 12 months (Table 3, Fig 2), with a gradual decline over time and no clear peak in recurrences. At the time of analysis, the median DFS had not been

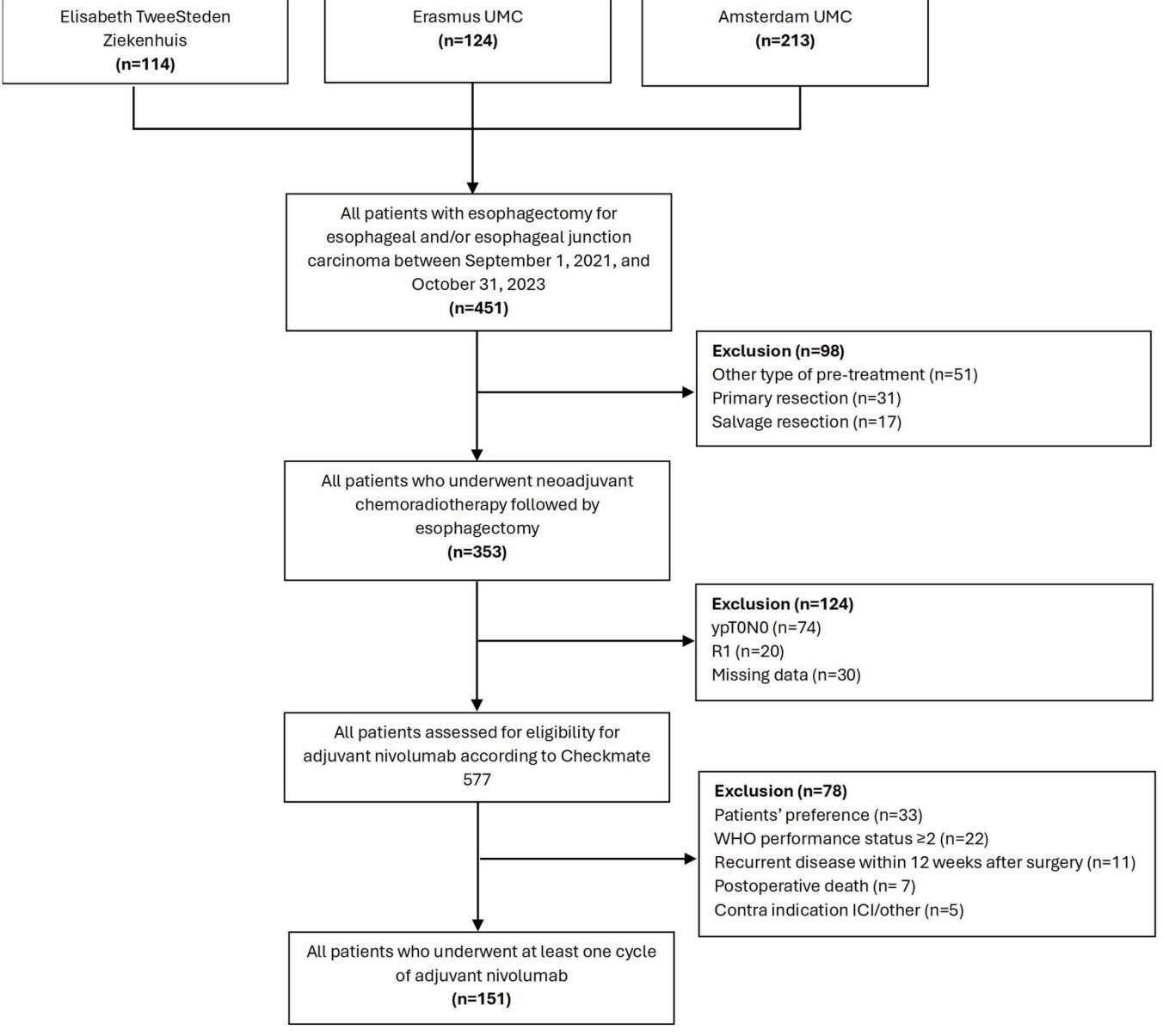

**Fig 1. Flowchart of patient selection.** Abbreviations: n; number, R1; microscopic irradical resection; UMC; University Medical Center, WHO; World Health Organisation, ypT0N0; pathological complete response.

reached due to limited number of events. The 12-month DFS in this cohort (75%) was higher than reported in the Checkmate 577 trial (Table 3).

### Method of disease detection

In 69 (46%) patients, a baseline CT scan was performed before the start of adjuvant nivolumab, with no recurrences detected. Median time between surgery and baseline CT was 10.6 weeks (IQR 7.9–13.0). After nivolumab initiation, the timing of routinely performed CT scans varied between every three months (Q3M, n = 67), or every four months (Q4M,

**Table 1. Baseline characteristics.**

|  | Nivolumab (n=151) |
|---|---|
| Median age (range) – year | 66 [60-72] |
| Male sex – no. (%) | 124 (82) |
| **Histological type – no. (%)** |  |
| Adenocarcinoma | 130 (86) |
| Squamous cell carcinoma | 15 (10) |
| Other or unknown | 6 (4) |
| **Disease stage at initial diagnosis – no. (%)*** |  |
| I | 27 (18) |
| II | 50 (33) |
| III | 70 (46) |
| Unknown | 4 (3) |
| **Type of resection – no. (%)** |  |
| Ivor Lewis esophagectomy | 118 (78) |
| McKeown esophagectomy | 24 (16) |
| Other | 9 (6) |
| **Pathological tumor status post-surgery – no. (%)*** |  |
| ypT0 | 9 (6) |
| ypT1 or ypT2 | 80 (53) |
| ypT3 or ypT4 | 62 (41) |
| **Pathological lymph-node status post-surgery – no. (%)*** |  |
| ypN0 | 66 (44) |
| ≥ypN1 | 85 (56) |
| R0 resection – no. (%) | 151 (100) |

*Pathological lymph-node status and tumor status are classified according to the criteria of the 8th edition of the Cancer Staging Manual of the American Joint Committee on Cancer [14]. Percentages may not total 100 because of rounding.

Abbreviations: no, number; ypT, tumor status post-surgery; ypN, lymph node status post surgery; R0, microscopically margin-negative resection.

n=84) according to local protocol. The diagnostic yield per routine CT evaluation was calculated as the number of detected recurrences divided by the number of routine CT scans at each timepoint. Denominators decrease over time due to prior recurrence, treatment discontinuation, or study end. In the Q3M regimen (n=67), 12 patients (18%) had a recurrence, with eight (67%) detected by routine CT. The diagnostic yield (i.e., detection of disease recurrence/all performed CT scans) of routinely performed CT scans was 5% (3/62) at 3 months, 8% (4/51) at 6 months, and 3% (1/32) at 9 months. At 12 months, 3/18 patients (17%) had recurrence on routine CT. In the Q4M regimen (n=84), 15 patients (18%) experienced on-treatment recurrent disease, with ten (67%) detected by routine CT. Diagnostic yield was 9% (6/68) at 4 months and 8% (4/49) at 8 months. No routine 12-month CT scans were performed in the Q4M group. In total, 9/27 (33%) recurrences were detected by symptom-driven testing (i.e., CT scan, gastroscopy).

### Predictors of on-treatment recurrent disease

To identify predictors of early (0–6 months) versus late (7–12 months) disease recurrence during treatment, an univariable sensitivity analysis was performed (Table 4). There were no significant differences between the groups in terms of age,

**Table 2. Treatment outcomes and details on disease recurrence.**

| | Nivolumab (n=151) |
|---|---|
| Completed number of cycles – median (IQR) | 10 (5 –13) |
| **Discontinuation of nivolumab, reason – no. (%)** | 76 (50) |
| Recurrent disease | 27 (36) |
| Immune related adverse events | 33 (43) |
| Patients' preference | 9 (12) |
| Other | 7 (9) |
| **Recurrent disease during treatment– no. (%)** | 27 (18) |
| Local recurrence | 1 (4) |
| Local regional recurrence | 2 (7) |
| Distant recurrence | 24 (89) |
| **Method of detection – no. (%)*** | |
| Evaluation CT scans | 18 (67) |
| Planned CT scans (symptoms) | 3 (11) |
| Other | 6 (22) |
| **Recurrent disease during follow-up – no. (%)** | 50 (33) |
| **Mortality** | 25 (17) |
| Cancer-related death | 21 (84) |
| Immune related adverse events | 1 (4) |
| Other or unknown | 3 (12) |

Percentages may not total 100 because of rounding.

Abbreviations: IQR, interquartile range; no, number; CT, computed tomography.

*Method of detection of disease recurrence during treatment.

**Table 3. Disease free survival and proportion of patients with recurrent disease in the nivolumab group, and the Checkmate 577 trial.**

| Treatment | Time (months from nivolumab initiation) | 1 | 2 | 3 | 4 | 5 | 6 | 7 | 8 | 9 | 10 | 11 | 12 |
|---|---|---|---|---|---|---|---|---|---|---|---|---|---|
| Nivolumab cohort | DFS | 99% | 97% | 93% | 89% | 89% | 87% | 85% | 84% | 80% | 78% | 78% | 75% |
| | No. recurrences | 2 | 3 | 5 | 6 | 0 | 4 | 3 | 1 | 5 | 3 | 1 | 3 |
| | No. distant metastasis | 2 | 3 | 4 | 6 | 0 | 3 | 3 | 0 | 5 | 2 | 1 | 2 |
| Checkmate 577$ | DFS | 98% | 96% | 84% | 81% | 79% | 72% | 71% | 70% | 67% | 67% | 65% | 62% |
| | DMFS | 99% | 97% | 88% | 85% | 84% | 78% | 78% | 75% | 73% | 72% | 70% | 68% |

Recurrence free survival is calculated in months from the date of start of nivolumab until local, local regionally or distant metastatic disease.

$patients in the Checkmate 577 trial who received nivolumab.

Abbreviations: DFS, disease free survival. DMFS, distant metastasis free survival.

sex, histological subtype, disease stage at diagnosis, or surgical technique. However, among patients who experienced on-treatment recurrence, both the early and late recurrence groups had a higher proportion of patients with ≥N1 disease post-surgery (p=0.021) compared to those without recurrence during treatment. Furthermore, in a subset of patients with available preoperative PD-L1 CPS data (n=41), those with early recurrence were significantly more likely to have a PD-L1 CPS<5 than patients without recurrence (p=0.001).

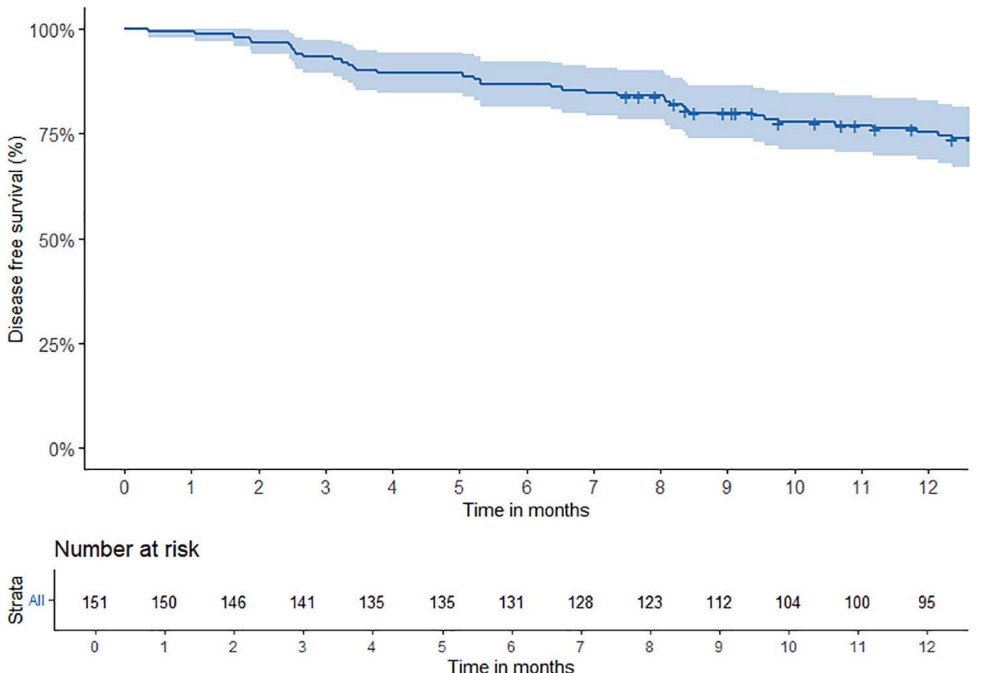

**Fig 2. Kaplan-Meier estimates of disease-free survival in patients who received adjuvant nivolumab.** Disease free survival is calculated from start date of nivolumab until recurrent disease or death, whichever occurred first.

### All-cause mortality

After completion or discontinuation of nivolumab, 23 (15%) patients developed recurrent disease within two years after initiation of adjuvant treatment. During a median follow-up of 21 months (IQR, 16–29), 25 patients (17%) died, with most common cause of death being cancer-related death (Table 2).

## Discussion

The introduction of adjuvant nivolumab in patients with residual disease after nCRT offers the potential for improved outcomes, as demonstrated in the Checkmate 577 trial. However, recurrent disease remains a significant challenge during nivolumab treatment, raising questions about the optimal follow-up strategy. Real-world data on follow-up strategies are lacking, leading to variability in CT scan intervals. While more frequent imaging may lead to earlier detection and prevent unnecessary exposure to ICI, it also increases healthcare burden and radiation exposure. This study includes data from three hospitals – two academic centers and one teaching hospital – reflecting a diverse patient population and increasing the generalizability of our findings. The variation in CT scan schedules between these institutions (every 3 months versus a 4- and 8-months regimen) allowed for a descriptive comparison of different follow-up regimens in a real-world setting.

A key finding is the limited diagnostic yield of baseline CT scans. Eleven patients (5%) had symptomatic recurrences diagnosed prior to nivolumab initiation, which is in line with previous studies reporting recurrences within three months after surgery in 7–15% of patients [20,21]. The median time from surgery to recurrence detection in these symptomatic patients was 7.9 weeks (IQR 4–10). For patients who underwent routine baseline CT scanning prior to nivolumab initiation (n = 69), the median time between surgery and baseline CT was 10.6 weeks (IQR 7.9–13.0), which largely overlapped with or was even later than the timing of symptomatic recurrences. This suggests that routine baseline CT would not have detected these recurrences earlier than clinical evaluation. Although early recurrences are common and could theoretically

**Table 4. Sensitivity analysis of predictors for on-treatment recurrent disease, stratified by time of recurrent disease (early recurrence 0-6 months, late recurrence 7-12 months and no on-treatment recurrence).**

| | Recurrent disease during nivolumab 0–6 months (n=21) | Recurrent disease during nivolumab 7–12 months (n=6) | No recurrence during active treatment (n=124) | *P value* |
|---|---|---|---|---|
| Median age (range) – year | 66 [61-72] | 63 [57-68] | 66 [60-72] | 0.544 |
| Male sex – no. (%) | 17 (81) | 5 (83) | 102 (82) | 0.987 |
| **Histological type – no. (%)** | | | | |
| Adenocarcinoma | 19 (91) | 5 (83) | 106 (86) | 0.813 |
| Squamous cell carcinoma | 0 | 0 | 15 (12) | 0.163 |
| **Disease stage – no. (%)** | | | | 0.589 |
| Stage I | 3 (14) | 0 (0) | 24 (19) | |
| Stage II | 5 (24) | 3 (50) | 42 (34) | |
| Stage III | 13 (62) | 3 (50) | 54 (44) | |
| **Type of resection – no. (%)** | | | | 0.225 |
| Ivor Lewis | 17 (81) | 3 (50) | 98 (79) | |
| McKeown | 3 (14) | 3 (50) | 18 (15) | |
| **PD-L1 status – no. (%)** | | | | |
| PD-L1 CPS<5[&] | 7/11 (33% of total early recurrences; 64% of those tested) | 1/1 (17) | 8/29 (7% of total group; 28% of those tested) | 0.001 |
| **Pathological tumor status post-surgery – no. (%)*** | | | | 0.172 |
| ypT0 | 1 (5) | 0 | 8 (7) | |
| ypT1 or ypT2 | 9 (43) | 1 (17) | 70 (57) | |
| ypT3 or ypT4 | 11 (52) | 5 (83) | 46 (37) | |
| **Pathological lymph-node status post-surgery – no. (%)*** | | | | 0.021 |
| ypN0 | 6 (29) | 0 | 60 (48) | |
| ≥ypN1 | 15 (71) | 6 (100) | 64 (52) | |

& PD-L1 expression was determined from tissue prior to neoadjuvant chemoradiotherapy (n=39) or resection specimen (n=2).

*Pathological lymph-node status and tumor status are classified according to the criteria of the eighth edition of the Cancer Staging Manual of the American Joint Committee on Cancer.

Percentages may not total 100 because of rounding. A two-sided P-value <0.05 was considered to indicate statistical significance.

Abbreviations: no, number; ypT, tumor status post-surgery; ypN, lymph node status post surgery; PD-L1, programmed death ligand 1; CPS, combined positive score.

be captured by baseline imaging, our data indicate that the baseline CT did not contribute to detection, as these events were already recognized based on clinical symptoms.

Regarding follow-up imaging intervals, our study showed variability across institutions. Both strategies showed nearly identical recurrence rates during treatment (18%) and rates of recurrence detection by routine CT (67%). While time-to-event analysis showed a relatively even distribution of recurrences, most were detected around months 4 and 8. However, as most patients were managed with the 4/8-month regimen, this may have introduced bias. The non-randomized design, differences between institutions, and the limited sample size hindered a meaningful comparison between the Q3 and Q4 month strategies.

Our sensitivity analyses did not identify clear clinical predictors of early (0–6 months) and late (7–12 months) on-treatment recurrence, with the exception of two observations. First, patients with a CPS<5 at diagnosis had a significantly

higher recurrence rate compared to those with a CPS ≥ 5. Second, the presence of one or more positive lymph nodes post-surgery was associated with early recurrence, consistent with prior studies linking higher postoperative nodal (ypN) status to disease recurrence [10]. However, since PD-L1 expression was not routinely assessed and the subgroup sizes were small, these findings should be interpreted with caution and requires validation in larger cohorts. In addition, differences in PD-L1 assessment methods—our study largely relied on available data from post-treatment samples, whereas CheckMate 577 used predominantly pre-treatment biopsies—complicate the interpretation of these observations. Further research is needed to confirm whether low PD-L1 expression and nodal positivity can serve as reliable biomarkers of early on-treatment recurrence, and to determine whether more frequent imaging strategies could facilitate earlier detection and improve patient outcomes. Heightened clinical awareness and close monitoring remain particularly important for these high-risk patients.

The 1-year DFS in our cohort was 75%, compared to 62% in the CheckMate 577 study. While no formal statistical tests were performed, these findings suggest that disease recurrence in this retrospective cohort may be similar to that observed in a large clinical trial. In the CheckMate 577, a peak in recurrences was observed at three months, coinciding with the first protocol-mandated CT scan at 12 weeks. This may explain the observed peak, although it is unclear how many recurrences were symptomatic versus detected through routine imaging. While we noted a slight increase in early recurrences between two and three months, our study showed a more evenly distributed recurrence rate over time, without a distinct critical time point for evaluation imaging, corresponding with daily practice as compared to a clinical trial cohort. Taken together, these data suggest that performing a CT scan around 3–4 months may be valuable for detecting early recurrences. Importantly, standardized imaging at 12 months, as in CheckMate 577, was not routinely performed in our cohort.

The therapeutic landscape for esophageal cancer is evolving, with perioperative FLOT (5-FU/leucovorin/oxaliplatin/docetaxel) increasingly used for adenocarcinoma following recent ESOPEC results [22]. This shift in treatment strategy alters the population eligible for adjuvant therapy following nCRT and surgery, as it now mainly reserved for patients with SCC, limited nodal involvement, or those unfit for FLOT. This change may affect the future applicability of current treatment protocols. The implications of these changes for the role of adjuvant nivolumab are not yet clear, as definitive evidence is still pending. While CheckMate 577 demonstrated improved disease-free survival, preliminary overall survival data did not show a statistically significant difference. Recent real-world studies have reported a potential benefit, although follow-up remains limited [23]. In this context, our study offers relevant data to inform follow-up strategies for patients currently receiving nCRT and adjuvant nivolumab.

Several limitations must be considered when interpreting the findings of this study. First, detection bias may have occurred, as some recurrences detected during scheduled imaging were already clinically suspected, potentially overestimating the detection rate in both strategies. Second, the non-randomized design and variability in institutional practices prevented a formal comparison between the Q3M and Q4M imaging strategies, as any differences between groups may reflect confounding factors rather than inherent differences. The generalizability of the results may be limited due to the homogeneity of the cohort, which was predominantly male (82%) and mainly comprised of adenocarcinoma patients (86%) from Dutch centers. These factors may limit the applicability of the findings to other populations, histologies, or healthcare settings. Lastly, while early detection may help spare patients from unnecessary treatments and their associated risks, timely detection of recurrence through routine imaging does not necessarily translate into better patient outcomes. Based on our data, we cannot determine whether detecting recurrence 1–2 months earlier with one strategy (Q3M vs Q4M) would lead to improved outcomes, such as greater opportunities for salvage surgery or other curative interventions, especially considering that the majority of recurrences was distant.

In conclusion, this retrospective cohort study showed that 18% of patients with esophageal cancer, previously treated with neoadjuvant chemoradiotherapy and resection, developed recurrent disease during adjuvant treatment with nivolumab. This observational study cannot determine the optimal imaging interval. Our findings describe current practice patterns and outcomes, which may inform clinical decision-making pending prospective comparative studies.

Routine baseline CT scans appear to have limited utility, as early recurrences were predominantly detected based on clinical symptoms. However, considering the timing of recurrences, follow-up intervals of 3–4 months for early recurrences and 7–8 months for late on-treatment recurrences could be reasonable options. Reducing the number of scans requires careful consideration of recurrence detection, imaging frequency, feasibility, and healthcare costs. At the same time, follow-up imaging remains essential, as many recurrences detected by routine CT scans were asymptomatic. Given the high cost of nivolumab and the potential for severe adverse effects, timely detection of recurrence is crucial to avoid unnecessary exposure to treatment in patients who may not benefit. The choice of routine imaging intervals should be guided by institutional resources, patient preferences, and clinical judgment.

## Supporting information

**S1 Checklist. STROBE checklist.**
(PDF)

## Acknowledgments

None

## Author contributions

**Conceptualization:** Tamara J. Huizer, Anniek Strijdhorst, Laurens V. Beerepoot, Leni van Doorn, Bas P. L. Wijnhoven, Bianca Mostert, Hanneke W. M. van Laarhoven.

**Formal analysis:** Tamara J. Huizer, Anniek Strijdhorst.

**Investigation:** Tamara J. Huizer, Anniek Strijdhorst.

**Methodology:** Tamara J. Huizer, Anniek Strijdhorst.

**Project administration:** Tamara J. Huizer, Anniek Strijdhorst.

**Resources:** Mark I. van Berge Henegouwen.

**Supervision:** Bianca Mostert.

**Visualization:** Hanneke W. M. van Laarhoven.

**Writing – original draft:** Tamara J. Huizer, Anniek Strijdhorst.

**Writing – review & editing:** Tamara J. Huizer, Anniek Strijdhorst, Laurens V. Beerepoot, Mark I. van Berge Henegouwen, Leni van Doorn, Sarah Derks, Bas P. L. Wijnhoven, Bianca Mostert, Hanneke W. M. van Laarhoven.

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
