## [Decision Letter · Decision Letter 0]

20 Feb 2026

PONE-D-25-61933Optimizing imaging strategies for adjuvant nivolumab in esophageal cancer: the planning of scanningPLOS One

Dear Dr. Strijdhorst,

Thank you for submitting your manuscript to PLOS ONE. After careful consideration, we feel that it has merit but does not fully meet PLOS ONE’s publication criteria as it currently stands. Therefore, we invite you to submit a revised version of the manuscript that addresses the points raised during the review process.

We look forward to receiving your revised manuscript.

Kind regards,

Zhanzhan Li

Academic Editor

PLOS One

**Journal Requirements:**

“The authors declare that they have no known competing financial interests or personal relationships that could have appeared to influence the work reported in this paper.

Disclaimer: TJH and AS have no discolures. HvL reports: Research funding and/or medication supply: Amphera, Anocca, Astellas, AstraZeneca, Beigene, Boehringer, BMS,Daiichy-Sankyo, Dragonfly, MSD, Myeloid, ORCA, Servier; Consultant/advisory role: Auristone, Incyte, Merck, Myeloid, Servier; Speaker role: Astellas, Beigene, Benecke, BMS, Daiichy-Sankyo, JAAP, Medtalks, Novartis, Springer, and Travel Congress Management B.V. BM reports: Research funding and/or medication supply: BMS, Pfizer; Consultant/advisory role: Lilly, AstraZeneca; Speaker role: Servier, BMS, Amgen. SD reports: a consultant or advisory role for BMS (related to checkpoint inhibitors); research funding, medication supply, or both from Incyte (related to checkpoint inhibitors); and speaker roles for Servier, BMS, and Benecke. LVB reports: speaker role: Medtalks, BMS, Servier, Travel Congress Management. BW reports: Research funding BMS, consulting and speaker fee Medtronic, speaker role Travel Congress Management B.V. MvBH declares consultancies for  Johnson and Johnson, Stryker, BBraun Intuitive and Medtronic. All fees and grants paid to institution.”

3. We note that your Data Availability Statement is currently as follows:

“All relevant data are within the manuscript and its Supporting Information files.”

Reviewers' comments:

Reviewer's Responses to Questions

**Comments to the Author**

1. Is the manuscript technically sound, and do the data support the conclusions?

Reviewer #1: Partly

Reviewer #2: Yes

Reviewer #3: No

2. Has the statistical analysis been performed appropriately and rigorously? 

Reviewer #1: Yes

Reviewer #2: Yes

Reviewer #3: No

3. Have the authors made all data underlying the findings in their manuscript fully available?

Reviewer #1: No

Reviewer #2: Yes

Reviewer #3: No

4. Is the manuscript presented in an intelligible fashion and written in standard English?

Reviewer #1: Yes

Reviewer #2: Yes

Reviewer #3: Yes

5. Review Comments to the Author

Reviewer #1: Dear Authors,

Thank you for submitting this manuscript evaluating CT imaging strategies during adjuvant nivolumab treatment for esophageal cancer. This study addresses a clinically relevant question that has practical implications for patient follow-up after curative-intent treatment. The multi-center design and real-world data complement the pivotal CheckMate 577 trial findings. I have carefully reviewed your manuscript against PLOS ONE publication criteria. Below I provide detailed feedback organized by major issues, minor issues, and specific recommendations for improvement.

MAJOR ISSUES

MAJOR ISSUE 1: DATA AVAILABILITY STATEMENT CONTRADICTION

There is a contradiction between your submission form responses and the manuscript text regarding data availability:

Submission Form (Page 4-5):

- "Yes - all data are fully available without restriction"

- "All relevant data are within the manuscript and its Supporting

Information files."

Manuscript Text (Lines 283-286):

- "Due to the nature of the study, in which exception consent was obtained from participants, the data used in this research cannot be made publicly available. Data sharing is not permitted as per the consent agreement."

RECOMMENDATION:

Option A - If data truly cannot be shared publicly:

1. Correct the submission form by selecting "No" for unrestricted availability

2. Revise the Data Availability Statement to read:

"Data cannot be shared publicly due to ethical restrictions related to participant consent. The ethics approval (MEC2023-0631) specified that individual-level data would not be publicly released. Requests for de-identified, aggregated data may be directed to the corresponding author (Anniek Strijdhorst, [email]) and will require approval from the Medical Research Ethical Committee of Erasmus University Medical Center Rotterdam."

3. Specify what data CAN be shared (e.g., aggregated statistics, additional subgroup analyses upon request)

Option B - If data can be made available:

1. Prepare a de-identified dataset removing all patient identifiers

2. Deposit in a public repository (e.g., Dryad, Figshare, or institutional repository)

3. Include the DOI or accession number in your Data Availability Statement

4. Ensure the manuscript text matches the submission form

I recommend Option A given your ethics approval conditions, but ensure all statements are consistent throughout the submission.

MAJOR ISSUE 2: CONCLUSIONS EXCEED WHAT THE DATA CAN SUPPORT

The manuscript title, abstract, and conclusions make claims that exceed what a retrospective, non-randomized observational study can establish:

Title: "Optimizing imaging strategies..."

Abstract (Lines 46-47): "Routine CT imaging every 4 months, starting at 4 months after surgery, effectively detects recurrences during adjuvant nivolumab treatment while reducing unnecessary imaging."

Conclusion (Lines 268-274): Implies the 4-month interval is the recommended or optimal strategy.

However, your data show that both imaging strategies produced equivalent outcomes:

Q3M group (n=67):

- 12 patients (17.9%) developed recurrence

- 8 of 12 recurrences (66.7%) detected by routine CT

- Diagnostic yield: 5% at 3 months, 8% at 6 months, 3% at 9 months

Q4M group (n=84):

- 15 patients (17.9%) developed recurrence

- 10 of 15 recurrences (66.7%) detected by routine CT

- Diagnostic yield: 9% at 4 months, 8% at 8 months

Both strategies showed nearly identical recurrence rates (~18%) and detection rates (~67%). Your study design cannot determine which is superior because:

1. Patients were not randomized to imaging intervals

2. Institutional practice determined interval (potential confounding)

3. Sample sizes preclude detecting meaningful differences

4. No formal statistical comparison between groups was performed

RECOMMENDATION:

1. Revise the title to:

"Characterizing imaging strategies for adjuvant nivolumab in esophageal cancer: a multi-center retrospective cohort study" OR "Imaging follow-up patterns during adjuvant nivolumab for esophageal

cancer: real-world data from three Dutch centers"

2. Revise the abstract conclusion (Lines 46-51) to:

"In this retrospective cohort, routine CT imaging at 4-month intervals detected the majority of on-treatment recurrences. The gradual decline in disease-free survival suggests that recurrences are distributed over

time rather than clustered at specific time points. These real-world data may help inform, but cannot definitively establish, optimal follow-up intervals. Prospective studies comparing imaging strategies

are needed."

3. Revise the Discussion conclusion (Lines 268-279) to explicitly state: "This observational study cannot determine the optimal imaging interval. Our findings describe current practice patterns and outcomes, which may inform clinical decision-making pending prospective comparative studies."

4. Add a statement acknowledging that both Q3M and Q4M showed equivalent outcomes in your cohort (12/67 vs 15/84 recurrences; 8/12 vs 10/15 detected by routine CT), and that selection of imaging interval should consider institutional resources, patient preferences, and clinical judgment rather than presumed superiority of one approach.

MAJOR ISSUE 3: MULTIPLE DATA DISCREPANCIES

Several numerical inconsistencies were identified that must be corrected:

DISCREPANCY 1 - Patient Age:

Line 160: "The nivolumab cohort included 151 patients, with a median age

of 69 years (IQR, 60-72)..."

Table 1: "Median age (range) - year: 66 [60-72]"

The IQR values match (60-72), but the median differs by 3 years (69 vs 66).

DISCREPANCY 2 - Lymph Node Status P-Value: Line 193-194: "both the early and late recurrence groups had a higher proportion of patients with ≥N1 disease post-surgery (p = 0.039)"

Table 4: Shows p = 0.021 for "Pathological lymph-node status post-surgery"

These p-values differ substantially (0.039 vs 0.021) for what appears to be the same analysis.

I suspect Table 1 (66 years) is correct for age based on the IQR, and Table 4 (p=0.021) may be correct for the lymph node analysis, with the text containing typographical errors - but please verify from original data.

MAJOR ISSUE 4: STROBE CHECKLIST NOT PROVIDED

Lines 100-101 state: "Data collection and reporting followed the Strengthening the Reporting of Observational Studies in Epidemiology (STROBE) guidelines."

However, no STROBE checklist is included with the submission. PLOS ONE recommends (and for some study types requires) completed reporting checklists as supplementary material.

RECOMMENDATION:

1. Complete the STROBE checklist for cohort studies (22 items) Available at: https://www.strobe-statement.org/checklists/

2. For each item, indicate:

- The page/line number where the item is addressed, OR - "N/A" with brief explanation if not applicable

3. Submit as "S1_STROBE_Checklist.pdf" or similar

4. Add a statement to Methods: "A completed STROBE checklist is provided as Supporting Information (S1 Checklist)."

MAJOR ISSUE 5: INCOMPLETE LIMITATIONS DISCUSSION

While the Discussion acknowledges some limitations (Lines 228-231, 238-243), several important limitations are not explicitly addressed:

1. INABILITY TO COMPARE IMAGING STRATEGIES: The non-randomized design means institutional preference determined imaging interval, not randomn assignment. Any differences (or lack thereof) could reflect selection bias or confounding rather than true equivalence.

2. LEAD-TIME BIAS: Earlier or more frequent detection of recurrence through imaging may not translate to improved outcomes. Patients detected earlier appear to survive longer from diagnosis simply because diagnosis occurred earlier, not because of any true survival benefit.

3. DETECTION BIAS: The 67% detection rate by routine CT may be influenced by the timing of scans. Some "symptom-driven" detections may have been caught on the next routine scan regardless.

4. LIMITED GENERALIZABILITY: The cohort is predominantly male (82%, 124/151), adenocarcinoma (86%, 130/151), from Dutch academic centers. Results may not generalize to other populations, histologies, or healthcare settings.

5. EVOLVING TREATMENT LANDSCAPE: With FLOT becoming standard for adenocarcinoma, the population eligible for nCRT + adjuvant nivolumab is changing, potentially limiting future applicability.

MINOR ISSUES

MINOR ISSUE 1: TYPOGRAPHICAL AND FORMATTING ERRORS

Location: Line 94-95

Issue: Awkward sentence structure

Current: "...an incomplete pathological response and received at least one cycle of adjuvant nivolumab."

Suggested revision: "...incomplete pathological response, and who received at least one cycle of adjuvant nivolumab."

Location: Throughout

Issue: Inconsistent spacing around punctuation and parentheses (For example line 66, 72 and 95)

Recommendation: Careful proofreading for formatting consistency

Cite line 68-69

MINOR ISSUE 2: PD-L1 ANALYSIS - CONFUSING PRESENTATION AND INTERPRETATION

The PD-L1 subgroup analysis in Table 4 has confusing presentation:

Table 4 shows for "PD-L1 CPS <5":

- Early recurrence (0-6 mo): 7/11 (33)

- Late recurrence (7-12 mo): 1/1 (17)

- No recurrence: 8/29 (7)

- P = 0.001

The format "7/11 (33)" appears to show:

- Numerator/Denominator with available PD-L1 data

- Percentage in parentheses refers to % of TOTAL group (7/21 = 33%)

This is confusing because:

- 7/11 = 63.6% of patients WITH PD-L1 data had CPS <5

- But (33) shows 33% of the total early recurrence group (7/21)

- Similarly, 8/29 = 27.6% but (7) shows 8/124 = 6.5%

The more clinically meaningful comparison is among patients WITH available

PD-L1 data:

- Early recurrence: 7/11 (63.6%) had CPS <5

- No recurrence: 8/29 (27.6%) had CPS <5

Additionally, this is based on very small numbers (only 41 patients had PD-L1 data available per Line 195), making the p=0.001 finding potentially unreliable.

RECOMMENDATION:

1. Clarify the Table 4 presentation - specify whether percentages refer to:

- Proportion of those with available PD-L1 data, OR

- Proportion of the total group

2. Consider presenting both:

"7/11 (63.6% of those tested; 33.3% of early recurrence group)"

3. More explicitly frame as hypothesis-generating in the text (Lines 195-196):

"In an exploratory analysis limited by small sample size (n=41 with available PD-L1 data), patients with early recurrence were more likely to have CPS <5 (7/11, 63.6%) compared to those without recurrence

(8/29, 27.6%), p=0.001. This finding should be interpreted with caution given the small numbers and requires validation in larger cohorts."

MINOR ISSUE 3: CT READING METHODOLOGY NOT SPECIFIED

The manuscript does not describe how CT scans were interpreted:

- Single radiologist or consensus reading?

- Were readers blinded to clinical information?

- Were standardized criteria used for defining recurrence on imaging?

RECOMMENDATION:

Add to Methods section (around Line 121-125):

"CT scans were interpreted by [board-certified radiologists / the treating institution's radiology department] according to standard clinical practice. Recurrence was defined as [new lesions suspicious for malignancy / RECIST criteria / clinical judgment]. [Readers were/were not blinded to clinical

symptoms at time of interpretation.]"

MINOR ISSUE 4: MISSING COMPARISON TABLE FOR SELECTION BIAS ASSESSMENT

Of 229 eligible patients, only 151 (66%) received nivolumab. The 78 patients who did not receive treatment may differ systematically from treated patients. Understanding these differences is important for assessing selection bias.

The reasons for not starting treatment are listed (Lines 155-157):

- Poor performance status: n=22

- Patient preference: n=33

- Contraindications: n=5

- Post-operative death: n=7

- Pre-treatment recurrence: n=11

However, baseline characteristics of these patients are not compared to treated patients.

RECOMMENDATION:

Add a supplementary table (S1 Table) comparing baseline characteristics

between:

- Patients who received nivolumab (n=151)

- Eligible patients who did not receive nivolumab (n=78)

Include: age, sex, histology, stage, pathological response, performance status, and reason for not receiving treatment. This allows readers to assess whether the treated cohort is representative of the broader eligible population.

MINOR ISSUE 5: TIMING CONVENTION INCONSISTENCY

Lines 117-118 define DFS from "start of adjuvant nivolumab," but some analyses and discussion reference time from surgery. This can cause confusion when interpreting timing of events.

For example:

- DFS is from nivolumab start (Line 117)

- But nivolumab starts ≤16 weeks after surgery (Line 117-118)

- Table 3 shows "Time (months)" without specifying reference point

- Discussion mentions "4 months after surgery" (Line 46)

RECOMMENDATION:

1. Clearly state the reference time point (surgery vs. nivolumab start) for each analysis

2. Consider adding a note that nivolumab was initiated within 16 weeks of surgery (per protocol), so timepoints approximately align

3. In Table 3 header, specify: "Time (months from nivolumab initiation)"

4. In abstract/conclusions, clarify whether "4 months after surgery" means

4 months from surgery or 4 months from nivolumab start

MINOR ISSUE 6: CONFIDENCE INTERVALS FOR KEY ESTIMATES

Key percentages are reported without confidence intervals:

- On-treatment recurrence: 27/151 (18%)

- Detection by routine CT: 18/27 (67%)

- 12-month DFS: 75%

This makes it difficult to assess precision of estimates, which is particularly important given the relatively small sample size.

RECOMMENDATION:

Add 95% confidence intervals for:

- On-treatment recurrence rate: 27/151 = 17.9% (95% CI: 12.1-24.9%)

- Detection by routine CT: 18/27 = 66.7% (95% CI: 46.0-83.5%)

- DFS at 4, 8, and 12 months (from Kaplan-Meier analysis)

MINOR ISSUE 7: REFERENCE TO UNPUBLISHED DATA

Lines 68-72 reference "Updated results from CheckMate 577, presented at the 2025 ASCO Annual Meeting" with overall survival data. Reference 12 cites this as a conference abstract.

RECOMMENDATION:

This is acceptable for providing context, but add a statement noting: "These data have been presented in abstract form only and have not yet undergone peer review. Final conclusions regarding overall survival benefit should await full publication."

MINOR ISSUE 8: DIAGNOSTIC YIELD CALCULATION CLARIFICATION

The diagnostic yield calculations (Lines 180-185) could be clearer:

Q3M group:

- 5% (3/62) at 3 months

- 8% (4/51) at 6 months

- 3% (1/32) at 9 months

- 17% (3/18) at 12 months

Q4M group:

- 9% (6/68) at 4 months

- 8% (4/49) at 8 months

The denominators (62, 51, 32, 18, 68, 49) represent patients who had CT scans at each timepoint, not the total cohort. This is appropriate but could be stated more explicitly.

RECOMMENDATION:

Add clarification: "Diagnostic yield was calculated as the number of recurrences detected divided by the number of patients who underwent routine CT scanning at each timepoint. Denominators decrease over time due to prior recurrence, treatment discontinuation, or study end."

QUESTIONS REQUIRING CLARIFICATION

1. PROTOCOL DEVIATIONS: Were there instances where patients deviated from their assigned imaging protocol (e.g., Q3M patient receiving scan at 4 months due to scheduling)? If so, how were these handled in the analysis?

2. SYMPTOM-DRIVEN VS. ROUTINE SCANS: How were "symptom-driven" scans (Table 2, n=3, 11%) distinguished from routine scans in your data collection? Was this based on documented clinical indication, or timing relative to protocol?

3. PRE-TREATMENT RECURRENCES: For the 11 patients with symptomatic recurrence before starting nivolumab, what was the median time from surgery to recurrence detection? This information would help contextualize the potential value of baseline imaging.

4. LOSS TO FOLLOW-UP: Were any patients lost to follow-up during the study period? If so, how many and how were they handled in survival analyses?

5. IMAGING PROTOCOLS: Were there differences in CT protocols between institutions (e.g., contrast, slice thickness, body regions imaged)? Could this affect detection sensitivity?

6. P-VALUE DISCREPANCY: Please clarify the correct p-value for the lymph node status analysis - is it 0.039 (Line 193-194) or 0.021 (Table 4)?

7. VERIFICATION: Given the multiple numerical discrepancies identified, can you confirm that all statistics have been independently verified against the source database?

STRENGTHS OF THE MANUSCRIPT

I want to acknowledge several strengths of this work:

1. CLINICAL RELEVANCE: This study addresses a genuine gap in clinical knowledge. The CheckMate 577 trial established efficacy but did not define optimal surveillance strategies. Clinicians need guidance on follow-up imaging, and real-world data are valuable.

2. MULTI-CENTER DESIGN: Including three hospitals (two academic, one teaching) enhances generalizability beyond single-institution experience. The variation in imaging protocols between centers (Q3M vs Q4M) provides a natural comparison, albeit non-randomized.

3. REAL-WORLD DATA: Complementing randomized trial data with real-world evidence is valuable for understanding how treatments perform in routine clinical practice outside controlled trial conditions.

4. TRANSPARENT REPORTING: The authors are forthcoming about limitations, including small sample size for subgroup analyses and lack of formal sample size calculation.

5. APPROPRIATE METHODS: The statistical approach (Kaplan-Meier, descriptive statistics) is appropriate for the study design and research questions.

6. CLINICAL CONTEXT: The Discussion appropriately situates findings withinthe evolving treatment landscape (FLOT, updated CheckMate 577 data).

7. PRACTICAL IMPLICATIONS: The finding that recurrences are distributed gradually over time (rather than clustered at a specific timepoint) has practical implications for follow-up scheduling.

8. COMPREHENSIVE DATA COLLECTION: The study collected relevant clinical variables including tumor characteristics, treatment details, imaging timing, and outcomes, enabling meaningful descriptive analysis.

FINAL RECOMMENDATION

RECOMMENDATION: MAJOR REVISION

This manuscript addresses an important clinical question and provides useful eal-world data on imaging surveillance during adjuvant nivolumab therapy for esophageal cancer. The multi-center design and appropriate methodology are strengths.

However, several issues must be addressed before publication:

CRITICAL (must be corrected):

- Resolve the data availability statement contradiction (Major Issue 1)

- Correct all numerical discrepancies - age (69 vs 66) and p-value

(0.039 vs 0.021) (Major Issue 3)

IMPORTANT (significantly affects interpretation):

- Temper conclusions to match observational study design; both Q3M and Q4M showed equivalent outcomes (Major Issue 2)

- Provide STROBE checklist as supplementary material (Major Issue 4)

- Expand limitations discussion to address lead-time bias, selection bias, and generalizability concerns (Major Issue 5)

MINOR (should be addressed but less critical):

- Fix typographical errors (Minor Issue 1)

- Clarify PD-L1 table presentation (Minor Issue 2)

- Add CT reading methodology (Minor Issue 3)

- Consider supplementary table comparing treated vs untreated (Minor Issue 4)

- Clarify timing conventions (Minor Issue 5)

- Add confidence intervals (Minor Issue 6)

- Add caveat about unpublished CheckMate 577 OS data (Minor Issue 7)

- Clarify diagnostic yield methodology (Minor Issue 8)

With appropriate revisions addressing these concerns, this work could make a meaningful contribution to clinical practice by informing (though not definitively establishing) follow-up imaging strategies for patients receiving adjuvant nivolumab after curative-intent treatment for esophageal cancer.

I encourage the authors to carefully verify all numerical data, temper their conclusions to match the observational study design, and resubmit.

Respectfully submitted,

Peer Reviewer

Reviewer #2: To the authors:

Thank you for submitting your manuscript entitled “Optimizing imaging strategies for adjuvant nivolumab in esophageal cancer: the planning of scanning,” which evaluates real‑world imaging strategies during adjuvant nivolumab therapy for esophageal or gastroesophageal junction cancer following neoadjuvant chemoradiotherapy (nCRT) and R0 resection. By comparing 3‑monthly versus 4‑monthly CT‑based follow‑up schedules and their respective diagnostic yields for recurrence detection, the study suggests that a 4‑monthly scanning interval may balance the need for timely detection with the aims of minimizing radiation exposure and controlling healthcare costs. Although this study is of considerable interest, there are several issues I would like to raise prior to publication.

1. Exclusions for nivolumab: The study reports that 74 patients (21%) with a complete pathological response (ypT0N0) and 20 patients (6%) with microscopic irradical resection (R1) after surgery were excluded from the nivolumab cohort. Could you further clarify the specific follow‑up strategies, including imaging intervals, implemented for these patients who did not receive adjuvant nivolumab? In particular, were follow‑up intervals extended for ypT0N0 patients or shortened for R1 patients, and what were the actual surveillance strategies used in routine practice for these groups?

2. Imaging strategy for high‑risk esophageal cancer patients: The study identifies ≥ypN1 disease and PD‑L1 CPS <5 as factors associated with a higher risk of on‑treatment recurrence, although PD‑L1 data were available only in a subset of patients. Could you clarify whether the proposed strategy of “routine CT imaging every 4 months” has been specifically evaluated in these higher‑risk subgroups? Alternatively, should a more frequent interval, such as a Q3M strategy, be considered for such patients to ensure sufficiently timely detection of recurrence?

3. Comparison of Q4M vs Q3M interval strategy based on the diagnostic yield of CT scans: The study reports that, in the Q3M group, the diagnostic yield for recurrence detection was 5% at 3 months, 8% at 6 months, 3% at 9 months, and 17% at 12 months. In the Q4M group, the diagnostic yield was 9% at 4 months and 8% at 8 months, and no routine 12‑month CT scans were performed. To definitively conclude that Q4M is “superior” to Q3M, the following questions need to be addressed:

3‑1. Rationale for the absence of 12‑month CT scans in the Q4M group: Please provide a clear explanation for why routine CT scans were not performed at 12 months for patients in the Q4M group. Understanding the rationale for this omission is important for a complete comparison of the longer‑term performance of the two imaging strategies.

3‑2. Statistical significance of differences in diagnostic yield: Given the observed diagnostic yields in the Q3M and Q4M groups, was any statistical analysis performed to assess whether there is a significant difference in recurrence detection rates between these schedules at comparable time points? A formal statistical comparison would strengthen any claims regarding the relative performance of the two strategies.

3‑3. Impact of delayed detection on patient outcomes: The Q4M schedule inherently introduces an approximately one‑month delay in the first routine evaluation (3 vs 4 months) and a two‑month delay in the second evaluation (6 vs 8 months) compared with the Q3M schedule. Could you discuss whether such delays might adversely affect patients’ chances for potentially curative or salvage therapy, or otherwise worsen prognosis? Timely identification of recurrence is important to initiate salvage treatment as soon as possible. Therefore, a thorough discussion of the clinical implications of these detection delays is warranted.

Reviewer #3: Conceptual issue:

The paper considers two protocols, Q3M which has CAT scans at 3, 6, 9 and 12

months, and Q4M which has CAT scans at 4 and 8 months. The authors would

like to reach some conclusion about whether Q4M is competitive with Q3M for

detection of relapse.

Unfortunately I think this is close to impossible due to the limitations of the

data. There is no shared timepoint between the two protocols, so at any time

where they could be compared, one has been scanned more recently than the other.

Relapses which are not caught by scan are sometimes caught because they become

symptomatic, so that later scans are not done at all, introducing further

complication.

The authors divide detected reccurrences into early (0-6 months) and late (7-12 months).

However, since Q4M cannot detect recurrences in months 5 or 6 until month 8,

an identical recurrence will be "early" in Q3M and "late" in Q4M. And the

"late" category does not include months 9-12 for Q4M at all as those can never

be caught by scanning, since no 12 month scan is done.

Perhaps there is a sophisticated type of statistical analysis which could work

here. This reviewer is not an expert in trial statistics. However, no

such analysis has been attempted.

The only hope I see of getting any answer to the relative effectiveness of

Q3M vs Q4M is Monte Carlo simulation where various assumptions can be made about

the underlying distribution of events, they can be subjected to Q3M and Q4M,

and the resulting simulated data can be compared to the real data.

The results of the study may be worth presenting anyway--there is a good deal

of information here. But I am very uncomfortable with almost all conclusions drawn

by the authors. Certainly the statement that Q4M seems adequate is not based

in the data.

Presentational issues:

(1) There are discrepancies both within the front matter (likely generated by

the submission website) and between front matter and paper. These need to be

checked and corrected. Specifically:

Front matter: states no funding

Paper: lists funding sources

Front matter: answers "yes" to "are data available?"; instructions specifically state

that "by request" should be treated as a "no" answer and explained

Paper: states that data are only available by request

Both paper and front matter state that there are no known competing financial

interests, and then go on to list, in detail, potential competing interests.

I don't understand this: it appears contradictory.

(2) The section "Cohort selection" is not sufficiently clear. I strongly

recommend making a flowchart. Classes of patients are mentioned without saying

whether they were included or excluded. For example, the text mentions

18 patients who were in other trials, and 74 patients who had a complete

pathological response. I think the 18 were included and the 74 were excluded

but this is not stated. It is also unclear throughout this section what the

denominator of the given percentages is. For example, 11 patients had

recurrence before adjuvant treatment: this is given as 5%, but 5% of what?

Total patients? Eligible patients? Why are these 11 separated from "the main

reasons for not starting adjuvant treatment"?--isn't recurrence a reason for

not starting adjuvant treatment?

The trial protocol does a better job with this; I recommend basing the publication

version on the protocol.

(3) A number of abbreviations are used without explanation: someone deeply

involved with this specific cancer will know them, but cancer researchers from

other subdisciplines may not, and general readers definitely will not. In one

case two different unexplained abbreviations turn out to refer to the same thing.

Minor points:

p. 8 "as advocated in the Checkmate 577" Strange choice of words: is "mandate"

or "recommend" intended?

p. 295 "discolures" for "disclosures"

p. 10 clarify whether CPS was based on specimen analysis previously carried out

by others (who? when? with what criteria?) or by the current researchers.

p. 10 Some patients apparently had recurrence before their first dose of

nivolumab (and were therefore not dosed). What was the DFS value for these patients?

Zero? Or were they excluded from DFS calculation?

p. 11 "were summarized as means with standard deviations or medians with

interquartile ranges" -- I am concerned this represents "experimenter degrees of

freedom." Why were two different approaches used?

p. 11 "real-world cohort and the CheckMate 577 trial" I would be comfortable

contrasting simulated data with real-world data, but this retrospective trial

and the prospective CheckMate were both done in the real world with real

patients. I recommend "retrospective" or simply "this study."

6. PLOS authors have the option to publish the peer review history of their article (what does this mean?). If published, this will include your full peer review and any attached files.

Reviewer #1: No

Reviewer #2: No

Reviewer #3: No

---

## [Author Response · Author response to Decision Letter 1]

23 Apr 2026

Point-to-point response letter

Reviewer 1 _______________________________________

Reviewer #1: Dear Authors,

Thank you for submitting this manuscript evaluating CT imaging strategies during adjuvant nivolumab treatment for esophageal cancer. This study addresses a clinically relevant question that has practical implications for patient follow-up after curative-intent treatment. The multi-center design and real-world data complement the pivotal CheckMate 577 trial findings. I have carefully reviewed your manuscript against PLOS ONE publication criteria. Below I provide detailed feedback organized by major issues, minor issues, and specific recommendations for improvement.

We would like to thank the Reviewer for the thorough and constructive comments. We have made every effort to address the recommendations and believe that these revisions have improved the manuscript. For clarity and readability, we have combined some points where appropriate. Please find below our responses to the Reviewer’s comments and a summary of the changes made to the manuscript.

MAJOR ISSUES

MAJOR ISSUE 1: DATA AVAILABILITY STATEMENT CONTRADICTION

There is a contradiction between your submission form responses and the manuscript text regarding data availability: Submission Form (Page 4-5) and information files." Manuscript Text (Lines 283-286):

"Due to the nature of the study, in which exception consent was obtained from participants, the data used in this research cannot be made publicly available. Data sharing is not permitted as per the consent agreement."

Thank you for your feedback regarding the data availability statement. We have updated the statement in the manuscript (page 12) as requested. However, we are unable to change the selection in the submission portal, as we do not have access to this option in the revision phase.

"Data cannot be shared publicly due to ethical restrictions related to participant consent. The ethics approval (MEC2023-0631) specified that individual-level data would not be publicly released. Requests for de identified, aggregated data (including summary statistics and additional subgroup analyses upon request) may be directed to the Department of Internal Oncology at Erasmus University Medical Center via interne.oncologie@erasmusmc.nl and will require approval from the Medical Research Ethical Committee of Erasmus University Medical Center Rotterdam."

MAJOR ISSUE 2: CONCLUSIONS EXCEED WHAT THE DATA CAN SUPPORT

The manuscript title, abstract, and conclusions make claims that exceed what a retrospective, non-randomized observational study can establish: Title: "Optimizing imaging strategies..." Abstract (Lines 46-47): "Routine CT imaging every 4 months, starting at 4 months after surgery, effectively detects recurrences during adjuvant nivolumab treatment while reducing unnecessary imaging." Conclusion (Lines 268-274): Implies the 4-month interval is the recommended or optimal strategy. However, your data show that both imaging strategies produced equivalent outcomes: Q3M group (n=67):

- 12 patients (17.9%) developed recurrence

- 8 of 12 recurrences (66.7%) detected by routine CT

- Diagnostic yield: 5% at 3 months, 8% at 6 months, 3% at 9 months

Q4M group (n=84):

- 15 patients (17.9%) developed recurrence

- 10 of 15 recurrences (66.7%) detected by routine CT

- Diagnostic yield: 9% at 4 months, 8% at 8 months

Both strategies showed nearly identical recurrence rates (~18%) and detection rates (~67%). Your study design cannot determine which is superior because:

1. Patients were not randomized to imaging intervals

2. Institutional practice determined interval (potential confounding)

3. Sample sizes preclude detecting meaningful differences

4. No formal statistical comparison between groups was performed

RECOMMENDATION:

1. Revise the title to:

"Characterizing imaging strategies for adjuvant nivolumab in esophageal cancer: a multi-center retrospective cohort study" OR "Imaging follow-up patterns during adjuvant nivolumab for esophageal

cancer: real-world data from three Dutch centers"

2. Revise the abstract conclusion (Lines 46-51) to:

"In this retrospective cohort, routine CT imaging at 4-month intervals detected the majority of on-treatment recurrences. The gradual decline in disease-free survival suggests that recurrences are distributed over

time rather than clustered at specific time points. These real-world data may help inform, but cannot definitively establish, optimal follow-up intervals. Prospective studies comparing imaging strategies

are needed."

3. Revise the Discussion conclusion (Lines 268-279) to explicitly state: "This observational study cannot determine the optimal imaging interval. Our findings describe current practice patterns and outcomes, which may inform clinical decision-making pending prospective comparative studies."

4. Add a statement acknowledging that both Q3M and Q4M showed equivalent outcomes in your cohort (12/67 vs 15/84 recurrences; 8/12 vs 10/15 detected by routine CT), and that selection of imaging interval should consider institutional resources, patient preferences, and clinical judgment rather than presumed superiority of one approach.

Thank you for your thoughtful and detailed feedback. We fully agree that, given the retrospective and non-randomized design of our study, we cannot definitively establish the superiority of any imaging regimen. We appreciate your careful consideration of the study’s limitations, including the lack of randomization, potential confounding by institutional practice, and limited sample sizes, which preclude formal statistical comparison between groups.

In response, we have revised the manuscript to ensure that our claims are closely aligned with the data and presented in a more descriptive manner. Specifically:

• The title and abstract have been updated to reflect the observational nature of the study and to avoid overstating the conclusions as following:

“Imaging strategies for follow-up during adjuvant nivolumab in esophageal cancer: a multicenter retrospective cohort study”

• The manuscript now explicitly states that both imaging strategies (Q3M and Q4M) produced nearly identical recurrence and detection rates, and that selection of imaging interval should be guided by institutional resources, patient preferences, and clinical judgment rather than presumed superiority. We revised the conclusion in the abstract as follows:

“Routine baseline CT imaging did not detect recurrences, while routine imaging during adjuvant nivolumab identified the majority of recurrences. The gradual decline in disease-free survival suggests that recurrences are evenly distributed over time, supporting a routine imaging interval, such as every 3 or 4 months as used in our study. These real-world data may help inform clinicians, and future studies can further evaluate optimal imaging intervals.”

• The discussion and conclusion sections have been revised to clarify that our findings describe real-world practice patterns and outcomes, which may inform clinical decision-making, but cannot determine the optimal imaging interval. We also acknowledge the need for prospective comparative studies to address this question.

“In conclusion, this retrospective cohort study showed that 18% of patients with esophageal cancer, previously treated with neoadjuvant chemoradiotherapy and resection, developed recurrent disease during adjuvant treatment with nivolumab. This observational study cannot determine the optimal imaging interval. Our findings describe current practice patterns and outcomes, which may inform clinical decision-making pending prospective comparative studies.

Routine baseline CT scans appear to have limited utility, as early recurrences were predominantly detected based on clinical symptoms. However, considering the timing of recurrences, follow-up intervals of 3–4 months for early recurrences and 7–8 months for late on-treatment recurrences could be reasonable options. Reducing the number of scans requires careful consideration of recurrence detection, imaging frequency, feasibility, and healthcare costs. At the same time, follow-up imaging remains essential, as many recurrences detected by routine CT scans were asymptomatic. Given the high costs of nivolumab and the potential for severe adverse effects, timely detection of recurrence is crucial to avoid unnecessary exposure to treatment in patients who may not benefit. The choice of routine imaging intervals should be guided by institutional resources, patient preferences, and clinical judgment.”

MAJOR ISSUE 3: MULTIPLE DATA DISCREPANCIES

Several numerical inconsistencies were identified that must be corrected:

DISCREPANCY 1 - Patient Age: Line 160: "The nivolumab cohort included 151 patients, with a median age of 69 years (IQR, 60-72)..." Table 1: "Median age (range) - year: 66 [60-72]". The IQR values match (60-72), but the median differs by 3 years (69 vs 66).

DISCREPANCY 2 - Lymph Node Status P-Value: Line 193-194: "both the early and late recurrence groups had a higher proportion of patients with ≥N1 disease post-surgery (p = 0.039)" Table 4: Shows p = 0.021 for "Pathological lymph-node status post-surgery" These p-values differ substantially (0.039 vs 0.021) for what appears to be the same analysis. I suspect Table 1 (66 years) is correct for age based on the IQR, and Table 4 (p=0.021) may be correct for the lymph node analysis, with the text containing typographical errors - but please verify from original data.

We apologize for the inconsistencies in the representation of the data. We have carefully reviewed the original dataset and corrected the manuscript as follows:

• Patient Age: The correct median age is 66 years (IQR 60–72), as shown in Table 1. The text has been updated to reflect this value.

• Lymph Node Status P-Value: The correct p-value for pathological lymph-node status post-surgery is 0.021, as indicated in Table 4. The text has been revised accordingly.

We have ensured that all numerical values in the manuscript are consistent with the original data. Thank you for bringing these discrepancies to our attention.

MAJOR ISSUE 4: STROBE CHECKLIST NOT PROVIDED

Lines 100-101 state: "Data collection and reporting followed the Strengthening the Reporting of Observational Studies in Epidemiology (STROBE) guidelines."

We apologize that the STROBE checklist was not readily found within the submission. It was originally uploaded as Supporting Information, which may have caused it to be overlooked among the supplemental files. To ensure clarity, we have re-uploaded the checklist as supplemental data and revised the manuscript as follows:

"Data collection and reporting followed the Strengthening the Reporting of Observational Studies in Epidemiology (STROBE) guidelines. (15) A completed STROBE checklist is provided as Supporting Information (S1 Checklist)."

MAJOR ISSUE 5: INCOMPLETE LIMITATIONS DISCUSSION

While the Discussion acknowledges some limitations (Lines 228-231, 238-243), several important limitations are not explicitly addressed:

1. INABILITY TO COMPARE IMAGING STRATEGIES: The non-randomized design means institutional preference determined imaging interval, not random assignment. Any differences (or lack thereof) could reflect selection bias or confounding rather than true equivalence.

2. LEAD-TIME BIAS: Earlier or more frequent detection of recurrence through imaging may not translate to improved outcomes. Patients detected earlier appear to survive longer from diagnosis simply because diagnosis occurred earlier, not because of any true survival benefit.

3. DETECTION BIAS: The 67% detection rate by routine CT may be influenced by the timing of scans. Some "symptom-driven" detections may have been caught on the next routine scan regardless.

4. LIMITED GENERALIZABILITY: The cohort is predominantly male (82%, 124/151), adenocarcinoma (86%, 130/151), from Dutch academic centers. Results may not generalize to other populations, histologies, or healthcare settings.

5. EVOLVING TREATMENT LANDSCAPE: With FLOT becoming standard for adenocarcinoma, the population eligible for nCRT + adjuvant nivolumab is changing, potentially limiting future applicability.

Thank you for these valuable suggestions. We have specifically addressed the issues of selection bias, lead-time bias, detection bias, limited generalizability, and the evolving treatment landscape, as suggested by the reviewer. We have extended the limitations section in the manuscript to address the points raised, as follows (page 10):

“Several limitations must be considered when interpreting the findings of this study. First, detection bias may have occurred, as some recurrences detected during scheduled imaging were already clinically suspected, potentially overestimating the detection rate in both strategies. Second, the non-randomized design and variability in institutional practices prevented a formal comparison between the Q3M and Q4M imaging strategies, as any differences between groups may reflect confounding factors rather than inherent differences. The generalizability of the results may be limited due to the homogeneity of the cohort, which was predominantly male (82%) and mainly comprised of adenocarcinoma patients (86%) from Dutch centers. These factors may limit the applicability of the findings to other populations, histologies, or healthcare settings. Lastly, while early detection may help spare patients from unnecessary treatments and their associated risks, timely detection of recurrence through routine imaging does not necessarily translate into better patient outcomes. Based on our data, we cannot determine whether detecting recurrence 1-2 months earlier with one strategy (Q3M vs Q4M) would lead to improved outcomes, such as greater opportunities for salvage surgery or other curative interventions, especially considering that the majority of recurrences were distant.”

MINOR ISSUES

MINOR ISSUE 2: PD-L1 ANALYSIS - CONFUSING PRESENTATION AND INTERPRETATION

The PD-L1 subgroup analysis in Table 4 has confusing presentation:

Table 4 shows for "PD-L1 CPS <5": The more clinically meaningful comparison is among patients WITH available

PD-L1 data: - Early recurrence: 7/11 (63.6%) had CPS <5 - No recurrence: 8/29 (27.6%) had CPS <5

Additionally, this is based on very small numbers (only 41 patients had PD-L1 data available per Line 195), making the p=0.001 finding potentially unreliable.

RECOMMENDATION:

1. Clarify the Table 4 presentation - specify whether percentages refer to:

- Proportion of those with available PD-L1 data, OR Proportion of the total group

2. Consider presenting both: "7/11 (63.6% of those tested; 33.3% of early recurrence group)"

3. More explicitly frame as hypothesis-generating in the text (Lines 195-196):

"In an exploratory analysis limited by small sample size (n=41 with available PD-L1 data), patients with early recurrence were more likely to have CPS <5 (7/11, 63.6%) compared to those without recurrence

(8/29, 27.6%), p=0.001. This finding should be interpreted with caution given the small numbers and requires validation in larger cohorts."

Thank you for this comment. We added these detailed numbers in Table 4, and added the more

cautious interpretation to the Discussion as requested.

MINOR ISSUE 3: CT READING METHODOLOGY NOT SPECIFIED

The manuscript does not describe how CT scans were interpreted: Single radiologist or consensus

reading? Were readers blinded to clinical information? Were standardized criteria used for defining recurrence

on imaging?

Thank you for this comment. We agree that the methodology for CT scan interpretation should be clarified. As the CT scans were performed as part of routine clinical care, they were interpreted by board-certified radiologists according to standard clinical practice at each participating center. No standardized research criteria or consensus readings were applied, and readers were not blinded to clinical information. We have clarified this in the revised manuscript as follows (Method section, page 4-5):

"CT scans were performed as part of routine clinical care and were retrospectively reviewed for the purposes of this study. Board-certified radi

---

## [Decision Letter · Decision Letter 1]

11 May 2026

Imaging strategies for follow-up during adjuvant nivolumab in esophageal cancer: a multicenter retrospective cohort study

PONE-D-25-61933R1

Dear Dr. Anniek Strijdhorst,

We’re pleased to inform you that your manuscript has been judged scientifically suitable for publication and will be formally accepted for publication once it meets all outstanding technical requirements.

Kind regards,

Zhanzhan Li

Academic Editor

PLOS One

Additional Editor Comments (optional):

Reviewers' comments:

Reviewer's Responses to Questions

**Comments to the Author**

1. If the authors have adequately addressed your comments raised in a previous round of review and you feel that this manuscript is now acceptable for publication, you may indicate that here to bypass the “Comments to the Author” section, enter your conflict of interest statement in the “Confidential to Editor” section, and submit your "Accept" recommendation.

Reviewer #1: All comments have been addressed

Reviewer #2: All comments have been addressed

2. Is the manuscript technically sound, and do the data support the conclusions?

Reviewer #1: Yes

Reviewer #2: Yes

3. Has the statistical analysis been performed appropriately and rigorously? 

Reviewer #1: Yes

Reviewer #2: Yes

4. Have the authors made all data underlying the findings in their manuscript fully available?

Reviewer #1: Yes

Reviewer #2: Yes

5. Is the manuscript presented in an intelligible fashion and written in standard English?

Reviewer #1: Yes

Reviewer #2: Yes

6. Review Comments to the Author

Reviewer #1: Dear Authors,

Thank you for the substantial and largely responsive revision. The

manuscript is significantly improved: the title and conclusions now match

the observational design, the prior numerical discrepancies are

reconciled, the Methods on CT interpretation and CPS provenance are now

specified, the limitations section has been broadened, the new patient-

selection flowchart (Figure 1) is a clear addition, and the STROBE

checklist is included as Supporting Information. The Reviewer 2 and

Reviewer 3 concerns about high-risk subgroups, the Monte Carlo idea, and

ambiguous cohort selection have been engaged thoughtfully. Below I

summarize the small number of new or residual issues that should be cleaned up before

acceptance.

REMAINING ISSUES (R1)

MAJOR ISSUES

MAJOR ISSUE 1 (R1): NEW NUMERICAL DISCREPANCY — FIGURE 1 VS RESULTS TEXT

PROBLEM:

The new flowchart (Figure 1) shows under Exclusion (n=78):

- Patients' preference (n=33)

- WHO performance status ≥2 (n=22)

- Recurrent disease within 12 weeks after surgery (n=10)

- Postoperative death (n=8)

- Contra indication ICI/other (n=5)

Total: 33 + 22 + 10 + 8 + 5 = 78 ✓

But the Results text (line 164-165) states:

"The main reasons for not starting adjuvant treatment were poor

performance status (n=22), patient preference (n=33), and symptomatic

recurrence (n=11) detected before starting adjuvant treatment."

And the Discussion (line 244-245) states:

"Eleven patients (5%) had symptomatic recurrences diagnosed prior to

nivolumab initiation."

The rebuttal letter also uses n=11.

The flowchart says 10, the text and rebuttal say 11. Likewise, the

flowchart shows postoperative death n=8, while the original R0

manuscript and rebuttal cited n=7.

Also, the text now lists only 22 + 33 + 11 = 66 patients out of 78

exclusions — leaving 12 unaccounted for in the narrative (the 5

contraindications and 7-8 postoperative deaths are silently dropped).

Given that the prior review flagged numerical inconsistencies as a major

issue, introducing a new mismatch between Figure 1 and the running text

undermines the verification statement provided in the rebuttal. It also

affects the median-time-to-recurrence figure (7.9 weeks IQR 4-10) since

the denominator differs.

SPECIFIC RECOMMENDATION:

1. Reconcile to a single number — verify whether n = 10 or n = 11

patients had pre-nivolumab symptomatic recurrence, and update Figure 1,

Results text (line 164), Discussion (line 244), and the rebuttal/flow-

chart consistently.

2. In the Results narrative, list ALL five exclusion reasons so the totals

reconcile to 78. Suggested wording:

"Reasons for not starting adjuvant treatment were patient preference

(n=33), poor performance status (WHO ≥2, n=22), symptomatic recurrence

detected within 12 weeks of surgery (n=10), postoperative death (n=8),

and contraindications to immune checkpoint inhibitors (n=5)."

3. Re-verify the 7.9-week median timing using whichever denominator is

correct.

MAJOR ISSUE 2 (R1): MISPLACED 95% CONFIDENCE INTERVAL

PROBLEM:

Results, line 175-176:

"During treatment, 27 (18%) (95% CI: 12-25%) patients developed

recurrent disease, predominantly distant metastases (89%). Eighteen

patients (12%) (95% CI: 46-84%) were asymptomatic and their recurrence

was detected by routine imaging."

The CI "46-84%" cannot describe a percentage of 12. It is the CI for

18/27 = 66.7% — i.e., the proportion of recurrences detected by routine

CT (which the original review estimated as 95% CI 46.0-83.5%). The CI

has been attached to the wrong proportion.

SPECIFIC RECOMMENDATION:

Rewrite the sentence so the CI is paired with the proportion it describes.

Suggested wording:

"During treatment, 27/151 patients (18%, 95% CI 12-25%) developed

recurrent disease, predominantly distant metastases (24/27, 89%). Of

these 27 recurrences, 18 (67%, 95% CI 46-84%) were detected by routine

imaging in asymptomatic patients."

Note: 18 detected by routine CT corresponds to Table 2 ("Evaluation CT

scans 18 (67)"), so the 18/27 framing is correct.

FIGURE ANALYSIS

F1 (Major) — Figure 1 (flowchart) vs Results text mismatch

Already covered under Major Issue 1 (R1). The flowchart itself is

well-constructed and the arithmetic within Figure 1 is internally

consistent. The mismatch is between the figure and the running text.

F2 (Minor) — Figure 1 wording: "Recurrent disease within 12 weeks after

surgery" is clearer than "symptomatic recurrence" (used in the text)

because it specifies the time window. Recommend the running text

adopt the same phrasing.

F3 (Minor) — Figure 1 abbreviation list: lists "WHO; World Health

Organisation" but the flowchart uses "WHO performance status ≥2."

Consider clarifying as "WHO performance status, World Health

Organization performance status scale" or "ECOG/WHO performance

status" so the figure stands alone.

F4 (Minor) — Figure 2 (Kaplan-Meier): The curve is appropriate. The

number-at-risk panel is included. The 12-month KM-estimated DFS

appears to land at ~75% with 95% CI ~67-82% (visual estimate). The

text reports 75% (95% CI: 69-83%) — the lower bound 69% is slightly

higher than the visual ~67%, but within plotting tolerance. No

action needed.

MINOR ISSUES

MINOR ISSUE 1 (R1): RESIDUAL TYPOGRAPHIC ERRORS

- Abstract, line 31 (and page 1 of rebuttal letter):

"DFSwas 89%" → "DFS was 89%" (missing space).

- Results, line 158: "78% received nCRT with carboplatin/paclitaxel.After

surgery" → add a space after the period.

- Methods, line 93 (clean MS): the parallelism is still broken —

"neoadjuvant treatment with chemoradiotherapy prior to surgery,

microscopic radical resection (R0), incomplete pathological response

and who received at least one cycle of adjuvant nivolumab." Suggest:

"...chemoradiotherapy prior to surgery, achieved microscopic radical

resection (R0) with incomplete pathological response, and received at

least one cycle of adjuvant nivolumab."

- Methods, lines 116-119 (tracked-change residue): in the tracked

version "biopies" and "surgerical" appear briefly; the clean version

uses "biopsies" and "surgical" — verify the typeset PDF uses the

clean spelling.

- Discussion, line 251 (clean MS): minor missing space —

"...clinical evaluation.Although..." → "clinical evaluation. Although"

- Conflict-of-interest section: "Daiichy-Sankyo" appears twice — the

brand is "Daiichi Sankyo." Recommend correcting.

- Methods, line 136-137: "summarized as mean with standard deviation

(SD) when normally distributed,or medians" — missing space after

comma.

MINOR ISSUE 2 (R1): LIMITATIONS — TWO ADDITIONS RECOMMENDED

The rewritten limitations paragraph (lines 287-301) is much improved but

two prior sub-points are not represented:

a) Selection bias for the 78 untreated patients: The reasons are listed

but the limitations paragraph does not acknowledge the residual

uncertainty this introduces. One sentence would suffice, e.g.:

"Although reasons for not starting nivolumab are reported, baseline

characteristics of the 78 untreated patients were not compared to the

treated cohort, leaving residual uncertainty regarding selection

effects."

b) Evolving treatment landscape: This is discussed in the body

(lines 275-285) but not flagged as a limitation. Consider adding:

"Finally, with perioperative FLOT becoming standard for

adenocarcinoma after the ESOPEC trial, the population eligible for

nCRT followed by adjuvant nivolumab is shifting, which may limit the

future applicability of these findings."

MINOR ISSUE 3 (R1): RESIDUAL DIRECTIONAL LANGUAGE IN DISCUSSION/CONCLUSION

The conclusion has been substantially tempered, but two phrases still

read as recommendations:

- Discussion, line 295-297: "Taken together, these data suggest that

performing a CT scan around 3-4 months may be valuable for detecting

early recurrences."

- Conclusion, lines 336-338: "...follow-up intervals of 3-4 months for

early recurrences and 7-8 months for late on-treatment recurrences

could be reasonable options."

These are appropriately hedged ("may be valuable," "could be reasonable

options") so this is not a barrier to acceptance, but for full

consistency with the new tempered framing, consider rewording the second

to:

"...the observed timing of recurrences was compatible with the 3-4 and

7-8 month intervals used in our cohort, but the optimal interval

cannot be inferred from this study."

MINOR ISSUE 4 (R1): COMPARISON WITH CHECKMATE 577 (75% vs 62% AT 12 MONTHS)

The Discussion (line 287-289) states:

"The 1-year DFS in our cohort was 75%, compared to 62% in the

CheckMate 577 study. While no formal statistical tests were performed,

these findings suggest that disease recurrence in this real-world

retrospective cohort may be similar to that observed in a large

clinical trial."

The reported difference (75% vs 62%) is a 13-percentage-point gap —

arguably better-than-CheckMate-577 outcomes, not "similar." Either the

language should reflect that the real-world cohort appears to do at

least as well as the trial, or possible explanations should be offered

(selection effects, different risk distribution, missed early recurrences

without standardized 12-week scan, shorter follow-up censoring effects,

higher proportion adenocarcinoma in real world, etc.).

STRENGTHS OF THE R1 REVISION

1. The title change ("Optimizing..." → "Imaging strategies for follow-up

a multicenter retrospective cohort study") cleanly resolves the

single biggest issue from R0.

2. The new Figure 1 flowchart is informative, internally consistent, and

clearly shows the 451 → 353 → 229 → 151 cascade across three centers.

3. Methods on CT interpretation, recurrence definition, symptom-driven

classification, and CPS provenance are now explicit and reproducible.

4. CIs added for headline estimates; CPS finding now correctly framed

as exploratory and hypothesis-generating.

5. Numerical discrepancies from R0 are corrected and verification was

independently performed.

6. Reviewer 3's conceptual concerns about non-comparability of Q3M and

Q4M are addressed by removing direct comparative statistical claims

and recasting the manuscript as descriptive.

7. The new clinical-relevance comparison of timing-from-surgery for

symptomatic recurrence (7.9 weeks) vs baseline CT (10.6 weeks) is a

nice quantitative addition that strengthens the "limited utility of

baseline CT" claim.

FINAL RECOMMENDATION

RECOMMENDATION: ACCEPT with MINOR REVISION

The R1 revision substantively addresses the prior major concerns

(overclaiming, data discrepancies, STROBE checklist, limitations,

data availability text, PD-L1 presentation, CT methodology, timing

convention, CIs, diagnostic-yield clarification). The new Figure 1 and

expanded Methods are real improvements.

What remains is a small set of clean-up items, none of which require

new analysis:

CRITICAL TO FIX:

- Reconcile Figure 1 vs text on n=10 vs n=11 pre-treatment recurrences

and n=7 vs n=8 postoperative deaths (Major Issue 1, R1).

- Move/relabel the misplaced 95% CI (46-84%) to the 67% (18/27) it

describes (Major Issue 2, R1).

EDITORIAL:

- Fix residual typos ("DFSwas," "paclitaxel.After," "Daiichy-Sankyo,"

awkward parallelism in the eligibility sentence) (Minor Issue 1, R1).

- Add one sentence each on selection bias (untreated cohort) and FLOT

evolving landscape to the limitations paragraph (Minor Issue 2, R1).

- Optionally soften the two remaining recommendation-flavored phrases

in Discussion/Conclusion (Minor Issue 3, R1).

- Reword the 75% vs 62% comparison so "similar" reflects the actual

13-point gap or offer explanations (Minor Issue 4, R1).

With these clean-up edits, the manuscript would be suitable for

publication.

Respectfully submitted,

Peer Reviewer

Review Date: May 2026 (R1)

Reviewer #2: Thank you for your thorough and thoughtful responses to my comments. All of my concerns have been adequately addressed, and the revisions have improved the clarity and rigor of the manuscript. I have no further questions or concerns at this time.

7. PLOS authors have the option to publish the peer review history of their article (what does this mean?). If published, this will include your full peer review and any attached files.

Reviewer #1: No

Reviewer #2: **Yes:** Ming-Ching Lee, MD, PhD, FACS, FCCP

---

## [Editor Report · Acceptance letter]

PONE-D-25-61933R1

PLOS One

Dear Dr. Strijdhorst,

I'm pleased to inform you that your manuscript has been deemed suitable for publication in PLOS One. Congratulations! Your manuscript is now being handed over to our production team.

Kind regards,

on behalf of

Dr. Zhanzhan Li

Academic Editor

PLOS One